# Encapsulation of Vitamin C by Glycerol-Derived Dendrimers, Their Interaction with Biomimetic Models of *Stratum corneum* and Their Cytotoxicity

**DOI:** 10.3390/molecules27228022

**Published:** 2022-11-18

**Authors:** Katia Bacha, Catherine Chemotti, Jean-Claude Monboisse, Anthony Robert, Aurélien L. Furlan, Willy Smeralda, Christian Damblon, Julien Estager, Sylvie Brassart-Pasco, Jean-Pierre Mbakidi, Jelena Pršić, Sandrine Bouquillon, Magali Deleu

**Affiliations:** 1Molecular Chemistry Reims Institute UMR CNRS 7312, Reims Champagne-Ardenne University, Boîte n° 44, B.P. 1039, F-51687 Reims, France; 2Laboratory of Molecular Biophysics at Interfaces (LBMI), Gembloux Agro-Bio Tech-University of Liege, Passage des Déportés, 2 B-5030 Gembloux, Belgium; 3Laboratoire de Biochimie Médicale et de Biologie Moléculaire, UMR CNRS/URCA 7369 (MEDyC), UFR Médecine, Reims Champagne Ardenne University, 51 Rue Cognacq Jay, F-51095 Reims, France; 4Structural Biological Chemistry Laboratory, MolSys Research Unity, University of Liege, 11, Allée du six Août, 4000 Liège, Belgium; 5Certech, Rue Jules Bordet, 45-Zone Industrielle C, B 7180 Seneffe, Belgium

**Keywords:** dendrimers, glycerol, vitamin C, encapsulation, membrane interactions, *stratum corneum* biomimetic membranes, liposomes, cytotoxicity

## Abstract

Vitamin C is one of the most sensitive cosmetic active ingredients. To avoid its degradation, its encapsulation into biobased carriers such as dendrimers is one alternative of interest. In this work, we wanted to evaluate the potential of two biobased glycerodendrimer families (GlyceroDendrimers-Poly(AmidoAmine) (GD-PAMAMs) or GlyceroDendrimers-Poly(Propylene Imine) (GD-PPIs)) as a vitamin C carrier for topical application. The higher encapsulation capacity of GD-PAMAM-3 compared to commercial PAMAM-3 and different GD-PPIs, and its absence of cytotoxicity towards dermal cells, make it a good candidate. Investigation of its mechanism of action was done by using two kinds of biomimetic models of *stratum corneum* (SC), lipid monolayers and liposomes. GD-PAMAM-3 and VitC@GD-PAMAM-3 (GD-PAMAM-3 with encapsulated vitamin C) can both interact with the lipid representatives of the SC lipid matrix, whichever pH is considered. However, only pH 5.0 is suggested to be favorable to release vitamin C into the SC matrix. Their binding to SC-biomimetic liposomes revealed only a slight effect on membrane permeability in accordance with the absence of cytotoxicity but an increase in membrane rigidity, suggesting a reinforcement of the SC barrier property. Globally, our results suggest that the dendrimer GD-PAMAM-3 could be an efficient carrier for cosmetic applications.

## 1. Introduction

The vectorization of active molecules is currently a major challenge in many research fields such as pharmaceuticals, food and cosmetics. In the pharmaceutical field, synthetic or bio-inspired nanocarriers are mainly developed to specifically deliver drugs such as anti-cancer agents [1,2,3,4,5] or therapeutic proteins [6]. In this area, genetically engineered organic, inorganic and virus-like nanostructures [7] or nanocarriers synthetized from ionic liquids [8] have increasing interest compared with conventional drug delivery systems. Some of them are particularly interesting for transdermal drug delivery [9,10,11].

The interest of nanovectorization for food [12,13,14] and cosmetic [15,16,17,18] sectors has also been increasing for several years. The vectorization allows for protection and delivery of active substances with low bioavailability, from their environment into the ones they are being introduced into by avoiding their metabolization or degradation along the way. In this context, symmetrical and branched macromolecules, called dendrimers, are one of the most promising vectorization systems [19].

Dendrimers are nanosized, hyper-branched, radially symmetrical molecules with a well-defined, homogeneous and monodisperse structure composed of branches that form internal tree-like cavities [20,21,22]. These compounds have many advantageous properties such as versatility, capacity to self-assemble and to participate in electrostatic interactions, chemical stability, low cytotoxicity and high aqueous solubility [23]. These specific properties are the key features making them advantageous for application in biomedical and industrial sectors [24,25]. In our previous works, we have developed a method for glycerol-based dendrimers synthesis and optimized their use as encapsulating agents of organic pollutants [26], contrast imaging agents [27], catalytic agents [28] and, recently, essentials oils [29,30].

In recent years, the favorable properties of dendrimers has aroused the interest of researchers for the optimization of production and application of these molecules and their implementation as a strategy in cosmetics. For instance, dendrimers are a good alternative for replacement of other conventional high molecular weight polymers in various cosmetic compositions [17,31,32]. Additionally, they also have a good potency to encapsulate sensitive cosmetical active substances in order to protect them and to ensure their release and a better efficiency [33,34,35].

Vitamin C is one of the most delicate cosmetic active ingredients and thus far there is no solution to limit its degradation in cosmetic products. Vitamin C is involved in many biological and physiological processes and is essential for normal body functioning [36,37]. It is necessary for the synthesis of collagen in the skin and for the healing of wounds [38]. It is also a powerful antioxidant and an anti-free radical agent that protects the epidermis from free radicals, thereby preventing oxidative damage and aging [39]. Nevertheless, vitamin C is a fragile compound and is easily decomposed into biologically inactive compounds by action of light, heat or oxygen [40]. To prevent this instability issue, different formulations have been developed for its vectorization and its controlled release. PPI and PAMAM dendrimers were tested to encapsulate vitamin C and other sensitive vitamins such as B3 and B6 [41]. Moreover, several kinds of water-soluble dendrimers have been used to encapsulate vitamin C in order to be incorporated into cosmetic or pharmaceutical products [33]. PPI dendrimers functionalized with polyethylene glycol entities seem to be suitable for the transport of vitamin C molecules in cosmetics and PAMAM dendrimers encapsulating vitamin C are used in dermatological applications [33,39,40,41,42].

In this work, after the synthesis of two families of glycerol-functionalized dendrimers, GD-PAMAMs and GD-PPIs, based on our previous works [26,27,28], we evaluated their capacity to encapsulate vitamin C and their cytotoxicity on human dermal fibroblasts. For the most promising candidate, GD-PAMAM-3, we further investigated its potential to deliver vitamin C into the uppermost layer of the skin by studying its interaction with biomimetic models of human SC using a biophysical approach. Two kinds of SC models were considered, lipid monolayers and liposomes. The first model was used to characterize the kinetics of adsorption and the interaction power of the dendrimer to the SC models, and the attractivity effect of the SC lipids on the dendrimer. The influence of dendrimers on SC fluidity and permeability [43] was characterized using SC mimicking liposomes that were preliminary optimized for these kinds of experiments. All lipid systems were composed of the three major lipid constituents of the SC extracellular lipid matrix (C24-ceramide-2, cholesterol and lignoceric acid) [44,45]. The effect of vitamin C was analyzed by comparing data on dendrimers alone and dendrimers encapsulating vitamin C. In addition, the influence of the pH on interaction and delivery properties was also evaluated.

## 2. Results and Discussion 

### 2.1. Encapsulation of Vitamin C

The ability of GD-PPIs and GD-PAMAMs to encapsulate vitamin C was evaluated by gravimetry after dialysis through a 1 kDa membrane, based on our previous works with organic pollutants [26]. Several encapsulation tests of vitamin C in GD-PPIs and GD-PAMAMs were performed in water at room temperature (Figure 1). The encapsulation results are summarized in Table 1.

As expected, the highest encapsulation is observed for the largest generations due to the multiplication of the peripheral groups and the increase in the number of cavities (Table 1, entry 3 versus entries 1 and 2). By comparing the two families derived from PPI and PAMAM (Table 1, entry 6 versus entries 1–4), we observed a better encapsulation of vitamin C with PAMAM-derived dendrimers. This is probably due to the presence of many amide functions and the presence of high amounts of heteroatoms, both of which favor electrostatic interactions. We also observed that glycerol-based PAMAM-3 encapsulates vitamin C better than commercial PAMAM-3. This is due to the presence of the hydroxyl groups in the molecule periphery. The encapsulation ability of GD-PAMAM-3 and GD-PPI-4 was supported by liquid-state nuclear magnetic resonance spectroscopy (NMR) experiments. In ^1^H NMR, a chemical shift variation was observed for the proton corresponding to the CH-O-CO of L-ascorbic acid in the presence of dendrimers (from 4.97 ppm for free vitamin C to 4.55 and 4.64 ppm for vitamin C encapsulated, respectively in GD-PAMAM-3 and GD-PPI-4) (Appendix A in Appendix A). This chemical shift can be due to an aggregation or an encapsulation of vitamin C. A 2D NOESY NMR experiment showed cross peaks between the protons of vitamin C and those of GD-PAMAM-3 and GD-PPI-4 dendrimers (CH_2_ in α of the carbamates and CH_2_ in α of the tertiary amines, respectively) (Appendix A in Appendix A). These peaks are caused by NOE interactions between protons that are in spatial proximity (distance lower than 5–6 Å). This last experiment confirms vitamin C encapsulation in GD-PAMAM-3 and GD-PPI-4.

Transverse relaxation time T_2_ measurements [46] were performed to confirm the interaction between vitamin C and GD-PAMAM-3. The conclusions were made based on transverse relaxation time values, since a small molecule binding to a larger one will have a shorter transverse relaxation time compared to when it is unbound. We observed a big difference in T_2_ between free vitamin C (T_2_ ≈ 1 s) and vitamin C encapsulated in GD-PAMAM-3 or vitamin C encapsulated in GD-PPI-4 (respectively T_2_ ≈ 0.3 s and T_2_ ≈ 0.2 s), confirming the interaction between vitamin C and GD-PAMAM-3 or GD-PPI-4 (Appendix A in Appendix A).

### 2.2. Cytotoxicity of Dendrimers

As the targeted application of our dendrimers is found in cosmetics, human dermal fibroblasts were used to evaluate their cytotoxicity. WST1 assays were performed to measure the viability of cells after dendrimer treatment at different concentrations. Cell staining with crystal violet assays allowing cell density measurements were carried out to confirm the results of the WST1 assay.

No cytotoxic effect was observed with GD-PAMAM-3 (from 0.001 µg/mL to 1 mg/mL) (Figure 1a), whereas the cell toxicity of dendrimers GD-PPI generations 3 and 4 was observed for concentrations equal or above 0.25 mg/mL. The viability percentage decreased with the increase of GD-PPI concentration (Figure 1b). The results show that GD-PPI-4 is less toxic than GD-PPI-3. This is probably due to the number of hydroxyl moieties present at the periphery (32 and 64 hydroxyl functions for GD-PPI-3 and GD-PPI-4, respectively). A higher number of hydroxyl groups favors electrostatic repulsion which limits the toxicity. Our results are in accordance with cytotoxicity data obtained previously with the same dendrimers on Human MRC5 fibroblasts from the lungs [25,26].

As GD-PAMAM-3 exhibits a high capacity to encapsulate vitamin C and presents weak cytotoxicity on human dermal fibroblasts, it is the best candidate for vitamin C delivery to the skin. To better understand its mechanism of action, its interaction with biomimetic models of SC was studied.

### 2.3. Interaction with Biomimetic Models of Stratum Corneum

Studies on the interaction of PAMAM-based dendrimers with skin using human [47,48,49] or animal skin tissues [50,51] showed that the concentration, charge and weight of PAMAM dendrimers influence both skin penetration and drug delivery [52,53]. Moreover, high generations of PAMAM dendrimers (above generation 4) caused pore formation [54]. It was also shown by fluorescence lifetime imaging microscopy that tecto-dendrimers penetrate the SC of human skin but do not cross the tight junction barrier [55]. In our work, we investigated the interaction of GD-PAMAM dendrimer with two SC biomimetic models: lipid monolayers and liposomes.

#### 2.3.1. Study on the Lipid Monolayer Model 

Lipid monolayers, the simplest membrane models, are used to study the ability of compounds to adsorb into the outer layer of membranes directly in contact with the external environment. In our study, lipid monolayers were made of *N*-Lignoceroyl-*D* sphingosine (C24Cer2), cholest-5-en-3-ol (Chol) and lignoceric acid (C24FA) (molar ratio 1:1:1), representatives of the major classes of lipids found in human SC. Three pH values (pH 5.0, pH 6.0 and pH 7.4) were considered for the subphase in order to mimic the different pH conditions of the skin that has an acidic pH in its external part and a gradual increasing pH reaching pH 7.4 in its inner part.

In the first step, the adsorption of the three components GD-PAMAM-3, vitamin C and the complex VitC@GD-PAMAM-3 was studied at a bare interface without lipids in order to evaluate their amphiphilic character at different pH values.

After injection of GD-PAMAM-3 into the subphase at pH 7.4, the surface pressure versus time curves increased and reached a plateau after 100 min (Figure 2a—blue curve), indicating that GD-PAMAM-3 is able to adsorb at the interface even if it is not as pronounced as conventional surfactants [56,57,58]. At the concentration of 0.08 µM of GD-PAMAM-3, the surface pressure reached its maximum (4 mN/m) and remained stable even at higher concentrations, revealing that GD-PAMAM-3 saturated the interface (Figure 2b). At pH 6.0, the maximum surface pressure at the equilibrium was 2 mN/m and at pH 5.0, there was no adsorption of the dendrimers at the air/water interface (data not shown), indicating that acid pHs are not favorable for the adsorption of the dendrimers.

Vitamin C did not adsorb at the interface even at a high concentration (11 µM) corresponding to its concentration in its encapsulated form and at any pH condition tested, likely because of its high hydrophilicity.

The complex VitC@GD-PAMAM-3 had the same adsorption behavior at a lipid-free interface as the dendrimers alone, regardless of the pH.

In the next step, the adsorption of the three components into the SC-mimicking lipid monolayer was monitored by measuring the surface pressure over time for different initial surface pressures of the monolayer in order to evaluate the interaction power of the dendrimer and the attractivity effect of the lipids at pH 7.4, pH 6.0 or pH 5.0. For the three pHs, an increase in the surface pressure over time is directly observed (example at pH 7.4 in Figure 2a). This indicates that the dendrimers are able to adsorb into the lipid monolayer. The time to reach the surface pressure plateau was much shorter (30–40 min) in the presence of the lipid monolayer than at the bare interface (>100 min). This suggests that SC- lipids promote the adsorption of GD-PAMAM-3 at the interface. From these data, the plot of the variation of the surface pressure at equilibrium (ΔΠ) following adsorption of GD-PAMAM-3 versus the initial surface pressure (Π_i_) of the SC mimicking monolayer was established (Figure 3).

A decrease of ΔΠ over Π_i_ increase is observed. The intersection of the linear regression with the *x*-axis allows for determining the maximum insertion pressure (MIP), which is related to the insertion power of the dendrimers in the lipid monolayer. Its intersection with the *y*-axis allows for determining the maximum surface pressure variation (ΔΠ_0_). By subtracting to ΔΠ_0_, the equilibrium surface pressure reached by the same concentration of GD-PAMAM-3 at a bare interface, the differential dΠ_0_ can be determined. It reflects the attractive effect of the lipids for dendrimers [57,59,60,61].

The values of MIP and dΠ_0_ for the three pHs are presented in Table 2. The MIP values of GD-PAMAM with or without vitamin C are close to 25 mN/m, which is in the range of the estimated surface pressure in native biological membranes. This suggests that dendrimers are able to penetrate into the lipid matrix of the SC. The pH does not influence the MIP values. The values of dΠ_0_ are positive at all pH values, indicating that the ternary mixture of lipids has an attractive effect on GD-PAMAM-3 dendrimers. This effect is accentuated at lower pH values (pH 5.0 and pH 6.0 compared to pH 7.4). At pH 6.0 and pH 7.4, there is no difference of dΠ_0_ values between GD-PAMAM-3 and VitC@GD-PAMAM-3, whereas at pH 5.0, a higher dΠ_0_ is observed for VitC@GD-PAMAM-3 than for GD-PAMAM-3. This indicates that at pH 5.0, encapsulated vitamin C increases the tendency of dendrimers to interact with the lipid monolayer. Additionally, it could suggest that vitamin C is released from the dendrimer and penetrates into the SC lipid monolayer. In skin conditions, vitamin C would thus be released in the external skin layers.

#### 2.3.2. Study on the Liposome Model 

In order to work with a biomimetic model that has physico-chemical features closer to those of cell membranes [62], liposomes mimicking the lipid composition of the SC had to be optimized preliminary. Two kinds of liposomes were developed, one adapted to the measurement of the SC permeability and another one suitable for analyzing the SC fluidity.


**
*Development and characterization of HPTS-DPX-loaded liposomes*
**


In order to perform permeability assays, the formulation of stable large uni-lamellar vesicles (LUVs) with the lipid composition C24Cer2/Chol/C24FA (1:1:1 molar ratio) and encapsulating a fluorescent probe and its quencher in their inner compartment was optimized. Formation of liposomes was possible at pH 7.4 but not at pH 5.0 and pH 6.0. At pH 7.4, one population of liposomes with a diameter of 180 ± 82 nm and that was stable over 5 days was obtained. An HPTS probe coupled with a DPX quencher (HPTS/DPX (8-hydroxypyrene-1,3,6-trisulfonic acid/p-xylene bis pyrimidium bromide)) was used, as it is the most appropriate fluorescent system for avoiding liposomes destabilization, which is observed with calcein-loaded liposomes (data now shown). In addition, the HPTS probe can be used at different pHs without any solubility concern. In order to better mimic real skin conditions, we ultimately formed liposomes with an internal core at pH 7.4 and an external environment at pH 5.0. For this purpose, a dialysis through a 20 kDa membrane in an acetate buffer at pH 5.0 was performed after formulation of the liposomes at pH 7.4. The change of the external pH did not affect the size and stability of the liposomes.


**
*Development and characterization of Laurdan-stained liposomes*
**


In order to measure the fluidity of the liposome bilayer, we formulated liposomes according to the protocol described in Section 3.8. After the formation of liposomes with a core at pH 7.4 and an outer environment at pH 5.0, Laurdan (1 µM) was added and incubated for 90 min at room temperature in the darkness. The size of these liposomes was 151 nm ± 42 nm and they were stable for 5 days.


**
*Effect of GD-PAMAM-3 on liposome permeability*
**


The percentage of liposome permeability was determined by measuring the leakage of the fluorescent probe and its quencher previously encapsulated and released by action of a given molecule. In this work, we performed leakage assays of HPTS-DPX-loaded liposomes upon treatment with three compounds, GD-PAMAM-3, vitamin C and VitC@GD-PAMAM-3.

The leakage upon the addition of GD-PAMAM-3 at T_0_ decreases with the decrease of GD-PAMAM-3:liposome molar ratio (Figure 4a). No permeabilizing effect at low molar ratios and limited permeabilizing effect (lower or equal 35%) at higher ratios was observed, which is in agreement with the low cytotoxicity observed on dermal cells. The permeabilizing effect appears upon the contact at T_0_ between the dendrimers and the liposomes and does not change over time (T 3h30). The small permeabilizing effect of GD-PAMAM-3 on the liposomes can be explained by the liposome aggregation phenomenon observed by cryo-TEM (see Appendix A in Appendix A), such as it was already detected for PAMAM-6 on other liposomes [63]. By comparing leakages induced upon GD-PAMAM-3 and VitC@GD-PAMAM-3 treatment, we noticed that there is no marked difference between the two (Figure 4a,b).

The permeability of the liposome by vitamin C was studied under the same conditions and the same concentration as it is in the VitC@GD-PAMAM-3 complex. There was no variation of fluorescence intensity neither with the increased concentration of vitamin C (from 0.0037 mM to 0.59 mM) nor with time (data not shown), indicating that vitamin C does not permeabilize the membrane.


**
*Effect of GD-PAMAM-3 on membrane fluidity*
**


The effect of GD-PAMAM-3 in absence or presence of vitamin C on the fluidity/rigidity of SC biomimetic liposomes was determined by measuring the variation of the generalized polarization (ΔGP) of Laurdan, staining the lipid bilayer, after the addition of the dendrimers (Figure 5). Namely, when the membrane phase (ordered/fluid) changes in the vicinity of the probe, the mobility of the lipid headgroup will change and will induce an emission spectral shift of Laurdan fluorescence [64]. Thus, before and after adding the dendrimers, the change of lipid order is monitored thanks to these spectral changes, which are expressed as the variation of the ΔGP. An increase of ΔGP corresponds to a rigidification of the bilayer.

Upon addition of GD-PAMAM-3 or VitC@GD-PAMAM-3 at a 1:2500 dendrimers:liposomes molar ratio, an immediate rigidifying effect of SC biomimetic liposomes was observed (ΔGP = 0.04 and 0.08, respectively) (Figure 5). This indicates an increase in bilayer rigidity, which is more pronounced in the presence of vitamin C. However, whereas an increase of ΔGP after 3h30 of incubation is observed with GD-PAMAM-3, a similar value is obtained for VitC@GD-PAMAM-3 at T0 and T3h30. It suggests that the presence of vitamin C accelerates the rigidity increasing process. It is important to note that when added alone, VitC has no effect on bilayer fluidity, likely because it does not reach the liposomes due its high hydrophilicity.

## 3. Materials and Methods

### 3.1. Chemicals

All generations of Poly(propylene imine) (PPI) were purchased from SyMOChem, all generations of Poly(amidoamine) (PAMAMs), glycerol carbonate, L-Ascorbic acid, Lignoceric acid (C24FA), 8-hydroxypyrene-1,3,6-trisulfonic acid (HPTS), *p*-xylene-bispyrimidium bromide (DPX), Triton-X100, Laurdan and Glutaraldehyde were purchased from Sigma-Aldrich (St. Louis, MI, USA). *N*-Lignoceroyl-D-sphingosine (C24Cer2) and cholest-5-en-3-ol (Chol) were purchased from Avanti Polar Lipids (Alabaster, AL, USA). Triethylamine (Et_3_N) was purchased from Fisher Scientific (Hampton, NH, USA). Tetrazolium salt (WST1) was purchased from Roche (Basel, Switzerland). 3-(Trimethylsilyl) propionic-2,2,3,3-d4 acid sodium salt (TMSP), deuterium oxide (D20) and methanol d4 (CD_3_OD) were purchased from Merck (Darmstadt, Germany).

All these chemical products were used without further purification. The buffer used in this work was a phosphate-buffered saline (PBS) comprised of sodium dihydrogen phosphate monohydrate (NaH_2_PO_4_·H_2_O)/disodium hydrogen phosphate (Na_2_HPO_4_)/sodium chloride (NaCl) (20/20/150 mM) at pH 7.4 and pH 6.0, an acetate buffer CH_3_COOH/CH_3_COO^−^ (5 mM) at pH 5.0 and an hydroxyethyl-piperazineethane-sulfonic acid buffer (HEPES) at pH = 7.6. The adjustment of the pH was done by adding an adequate amount of 3 M sodium hydroxide (NaOH).

### 3.2. Synthesis of Glycerol Derived Dendrimers 

The synthesis of glycerol-functionalized PAMAMs (glyceroDendrimer Poly(AmidoAmine) (GD-PAMAMs)) and glycerol-functionalized PPIs (glyceroDendrimer Poly(PropyleneImines) (GD-PPIs)) was performed as previously described [26,28]. It was done by reaction of commercial PPIs (generations 1–4) and PAMAMs (generations 1–3) with glycerol carbonate, (CG) in methanol/ethanol in the presence of Et_3_N (Figure 2a,b). To a bicol flask containing 1 eq of PPIs or PAMAMs dissolved in Methanol (MeOH) or (Ethanol) EtOH (20 mL), glycerol carbonate (2.5 equiv per NH_2_) and triethylamine (2.5 equiv per NH_2_) were added. The reaction mixture was refluxed under an inert atmosphere for 20 h. The solvent was evaporated under reduced pressure. The crude product was dissolved in a minimum of MeOH and then precipitated 3 times with an excess of a 2:1 (*v*/*v*) pentane/Ethyl acetate (AcOEt) mixture. The residue was dried under reduced pressure then dissolved again in a minimum of distilled water and purified by dialysis (cutoff 1 KDa) several times in distilled water. The product was obtained in a form of a yellow wax. NMR spectra were recorded at 298 K at 500 MHz for ^1^H and 125 MHz for ^13^C on an Avance NEO Bruker spectrometer equipped with a SP BB&19F/1H Iprobe CD_3_OD or D_2_O. The synthesis results are presented in the Table 3.

### 3.3. Encapsulation Procedure

GD-PPIs and GD-PAMAMs (1 equiv.) were dissolved in H_2_O (5 mL). Then, L-ascorbic acid (100 equiv.) was added, and the mixture was stirred 24 h at room temperature. The crude mixture was purified by dialysis (cutoff 1 KDa) three times (3 × 40 min) against distilled water (1000 mL) to eliminate the vitamin C excess. Finally, the water was removed by lyophilization to obtain a yellow wax.

NMR spectra were recorded at 298 K at 500 MHz for ^1^H and 125 MHz for ^13^C on an Avance NEO Bruker spectrometer equipped with a SP BB&19F/1H Iprobe in D_2_O.

### 3.4. NMR T_1_ and T_2_ Measurement

Longitudinal relaxation time T_1_ was determined with inversion-recovery pulse sequence (Appendix A in Appendix A). The delay between the inversion pulse and the read pulse and the relaxation delay were adjusted depending on its values (relaxation delay should be with a minimum 5 times the highest T_1_ in the sample). The number of scans was set at 8. Transversal relaxation time T_2_ was determined by the Carr-Purcell-Meiboom-Gill (CPMG) pulse sequence. The delay between 180 degrees pulse in the CPMG block was 2 ms for the dendrimers with vitamin C and 1.4 ms for free vitamin C. The number of loops and the relaxation delay were adjusted depending on the values of relaxation time. The number of scans was set at 8.

### 3.5. Cytotoxicity Assays

Cytotoxicity assays of the dendrimers were performed on human dermal fibroblast cultures purchased from PromoCell. Two different methods were used: (i) Reduction of a tetrazolium salt (WST1) by mitochondrial oxido-reductases and (ii) staining of cells with crystal violet. Human dermal fibroblasts were cultivated at a confluence in 75 cm^2^ culture flasks (Falcon) in DMEM medium containing glutamine (2 mM) and Fetal Calf Serum (FCS) at 10% (*v*/*v*) under a humid atmosphere with 5% CO_2_ and at 37 °C. Confluent human dermal fibroblasts (20,000 cells/mL) were deposited in 24 well-culture plates (Falcon) and cultured in the same conditions above. Culture medium is removed and replaced by the same DMEM medium with FCS 0.5% for 24 h. The different dendrimers were dissolved in distilled water and tested at various concentrations (ranging from 0.0001 to 1 mg/mL) for 48 h in the same DMEM medium with 0.5% FCS (1 mL/well).

WST1 reduction: At the end of the incubation period in the presence of different products, cells were rinsed with 1 mL of DMEM containing 0.5% FCS, and incubated with 400 µL of the same DMEM medium containing 10 µL of a WST1 solution in 400 µL, for 30 min at 37 °C. Tetrazolium salts were cleaved to formazan by the succinate-tetrazolium reductase system (EC 1.3.99.1), which belongs to the respiratory chain of the mitochondria, and is only active in metabolically intact cells. Volume of 100 µL of cell solutions from different treatments tested was transferred in 96 well-plates and absorbance at 450 nm was recorded with a spectrophotometer.

Cell staining with crystal violet allows to measure the cell density. After incubation with WST1, cells were washed with 1 mL of Phosphate Buffered Saline (PBS), then fixed by incubation for 20 min with 1 mL/well of PBS containing glutaraldehyde 1.1% (*v*/*v*). After abundant rinsing with distilled water, cells were stained by addition of 400 µL/well of a solution of crystal in a HEPES buffer at pH = 7.6 for 20 min. The excess of crystal violet was removed, and cells were extensively rinsed with distilled water. Staining was extracted with 1 mL of acetic acid 10% (*v*/*v*). Volumes of 100 µL of the colored solution were transferred onto 96 well-plates. Absorbance at 560 nm was recorded with a spectrophotometer.

Results are expressed as the mean ± standard deviation. Each experiment is done in quadruplicate.

### 3.6. Study on Lipid Monolayer Models by Langmuir Film Balance Technique

The experiments were performed at 22 ± 1 °C with an automated LB system (KSV minitrough, KSV instruments Ltd., Helsinki, Finland). The adsorption area (75 × 160 mm^2^) is delimited by two symmetrical barriers held stationary at the edges of the trough for the duration of the measurement. For the study of adsorption in the absence of lipids, solutions containing the vitamin C, GD-PAMAM-3 or VitC@GD-PAMAM-3 (GD-PAMAM-3 with encapsulated vitamin C) were prepared in ultra-pure water and then injected into the PBS at pH 7.4 or 6.0 or an acetate subphase at pH 5.0 at different concentrations (from 1 µM to 50 µM), (from 0.025 µM to 5 µM) and (11.33 µM), respectively, for Vitamin C, GD-PAMAM-3 and VitC@GD-PAMAM-3), using a Hamilton syringe and two custom-made devices allowing the injection of products without disturbing the air–water interface. These devices were placed at two fixed positions on the trough to ensure a reproducible injection process. Furthermore, during the entire duration of the experiment, the subphase was stirred using two cylindrical micromagnetic rods (8 × 1.5 mm^2^) and two electronic stirrer heads located beneath the trough (model 300, Rank Brothers, Bottisham, UK). An auto-reversing mode with slow acceleration and a stirring speed of 100 rpm were selected. After the injection of the products, the increase of the surface pressure was measured by means of a Wilhelmy plate connected to the measuring system of the device. The surface pressure was recorded as a function of the time until reaching a plateau.

For the study of the adsorption in the presence of lipids, a precise volume of lipid mixture (C24Cer2/Chol/C24FA) (1:1:1), prepared in chloroform/methanol (2:1 *v*/*v*), was spread on the surface of the buffer until the desired initial surface pressure. As soon as the initial surface pressure was stabilized (20 min to 45 min), solutions of vitamin C, GD-PAMAM-3 or VitC@GD-PAMAM-3 (20 µL) were injected into the buffer subphase at final concentrations of 11.13, 0.2 and 11.33 µM and the increase in surface pressure was recorded as a function of time.

### 3.7. Formulation of Fluorescent Probe-Encapsulated Liposomes

In a round bottom flask, the three lipids, C24Cer2, Chol and C24FA were weighted in equimolar amounts. The lipids were solubilized in chloroform/methanol (2/1, *v*/*v*) and the solvent removed under reduced pressure. The resulting lipid film was then dried in a vacuum desiccator overnight to remove the remaining solvent. The lipid film was then rehydrated with a PBS buffer at pH 7.4 (1 mL) containing the fluorescent probe, 8-hydroxypyrene-1,3,6 trisulfonic acid (HPTS 6 mM) and its quencher, p-xylene-bis-pyridinium bromide (DPX 10 mM), previously heated to 90 °C to obtain an initial concentration of 3 mM of liposomes. The flask was vortexed to solubilize the lipid film and was placed in a water bath at 90 °C, which corresponds to a temperature above the lipid transition temperature. The mixture was vortexed every 10 min for one hour, and the content of the flask was transferred to a polypropylene tube. Seven cycles of freezing (in liquid nitrogen) and heating (in a water bath at 90 °C) were then performed. The heat shock allows the formation of homogeneous multi-lamellar vesicles (MLVs). MLVs are extruded 21 times through polycarbonate filters with a pore diameter of 100 nm to obtain large uni-lamellar vesicles (LUVs). A Sephadex-G75 gel purification step was performed to remove the unencapsulated probe. The Sephadex gel was obtained by hydrating 1.7 g of Sephadex G75 with 60 mL of PBS buffer at pH 7.4 at 150 °C for 2 h. Finally, a dialysis with a 20 kDa membrane in 500 mL of acetate buffer at pH 5.0 was performed for 1 h in order to equilibrate the pH value, inside and outside the dialysis membrane. The objective of this step is to form liposomes with a core at pH 7.4 and an external environment at pH 5.0 to better mimic the real conditions of the skin. The liposomes were then stored in the dark at 40 °C. The concentration of HPTS/DPX-loaded liposomes has been determined by NMR analysis.

### 3.8. Formulation of Laurdan-Stained Liposomes

The lipid film (C24Cer2/Chol/C24FA) (1:1:1), formed by the method described above, was rehydrated with a PBS buffer at pH 7.4, previously heated to 90 °C to obtain an initial concentration of 3 mM of liposomes. The flask was placed in a water bath at 90 °C. The next steps were carried out in the same way as described for fluorescent probe-encapsulated liposomes, except the Sephadex gel purification step was not performed. After liposomes formation, a solution of Laurdan at 1 mM was added to the liposome solution to obtain a solution with 1 µM of Laurdan, which was incubated for 90 min at room temperature in the darkness. The Laurdan solution (1 mM) was prepared by adding 0.35 mg of Laurdan in 1 mL of DMSO.

### 3.9. Liposomes Characterization: Size and Quantification

#### 3.9.1. Size Determination—DLS Measurement

Liposome sizes were determined by Dynamic Light Scattering with a Zetasizer Nano ZS apparatus (Malvern Instrument SA, Worcestershire, UK) equipped with a laser emitting at 632 nm. Acquisitions were measured at 25 °C (equilibration duration of 15 min) using a lipid concentration of 1.5 mM. For each sample, from 10 to 20 acquisitions were carried out with an acquisition time of 10 s. Autocorrelation functions obtained were fitted with a CONTIN algorithm as provided in the software in order to obtain the precise values of the liposomes’ diameter and size distribution. In all cases, measurements were repeated three times.

#### 3.9.2. Morphology Characterization—Cryo-TEM

The samples were prepared in the same way as described in the Laurdan-stained liposomes preparation without adding Laurdan.

A 4-μL aliquot of sample was deposited on an EM grid coated with a perforated carbon film. After draining the excess liquid with a filter paper, grids were plunge-frozen into liquid ethane cooled by liquid nitrogen using a Leica EMCPC cryo-chamber (Leica Microsystems, Wetzlar, Germany). For cryo-TEM observation, grids were mounted onto a Gatan 626 cryoholder and transferred to a Tecnai F20 microscope (Thermo Fisher Scientific) operated at 200 kV. Images were recorded with an Eagle 2k CCD camera (FEI, Hillsboro, OR, USA).

#### 3.9.3. Lipid Concentration Quantification—NMR Spectroscopy

The lipid concentration of the LUVs was quantified by ^1^H NMR spectroscopy following a protocol from Hein et al. (2016) [65] and Smeralda et al. (2019) [66]. TMSP was used as an internal standard (δ = 0 ppm). ^1^H NMR measurements were performed at 50 °C on a Bruker Avance III HD spectrometer (Bruker) equipped with a 5 mm TCI CryoProbe operating at 700 MHz. The large H_2_O signal was attenuated thanks to a pulse program with presaturation using a composite pulse (zgcppr). The instrument’s standard settings were: 90° pulse angle, 1.47 s acquisition time, 2 s relaxation delay and 16 ppm spectral width. Locking and shimming were performed on the signal of the exchangeable deuterium atoms of the D_2_O/CD_3_OD mixtures. In total, 128 scans were performed leading to a total acquisition time of 7 min 30 s. Data processing was performed with Topspin 3.5. After zerofilling to 64 k data points, apodization (exponential and Gaussian functions, 1 Hz), Fourier transformation, phase- and baseline correction and the peak areas were determined by integration.

### 3.10. Leakage Assays

The measurements were performed on a PerkinElmer–LS 55 fluorometer, at an excitation wavelength of 412 nm and an emission wavelength of 513 nm with a slit value around three for both. In an aluminum-coated Falcon, a solution of liposomes enclosing the fluorescent probe was prepared at 15 µM. In addition, a 1% TritonX-100 solution was prepared. A cuvette of 10 µL of solvent (ultra-pure water) was added to 750 µL of the liposome solution and the content of the cuvette was gently mixed to homogenize the content without forming air bubbles. The fluorescence intensity measurement was performed and corresponds to F_0_. Keeping the same parameters used for the first measurement (slit value), a measurement with the addition of 10 µL of 1% Triton-X100 in 750 µL of the liposome solution was performed. The intensity value obtained corresponds to the F_Triton_ value. Similarly, measurements were performed for GD-PAMAM-3, VitC@GD-PAMAM-3 (56:1, VitC: dendrimer molar ratio) and vitamin C samples for a range of concentrations by adding 10 µL of the sample to 750 µL of the liposome solution. The values obtained correspond to the F_sample_. The absence of fluorescence emitted by the sample was controlled by adding 10 µL of the sample to 750 µL of buffer.

At higher dendrimers:liposomes molar ratios, negative values were observed and were attributed to the capacity of the dendrimers to re-encapsulate the HPTS/DPX released from the liposomes. The percentage of re-encapsulated HPTS/DPX was determined in independent experiments performed at the same dendrimer concentrations as in initial experiments and with a concentration of HPTS/DPX corresponding to the fluorescence intensity obtained in the initial experiments in the presence of Triton X-100. For each dendrimer:liposome molar ratio, the value of the re-encapsulated HPTS/DPX percentage was added to the initial value of leakage to obtain the final Leakage (%) according to Equation (1): Leakage (%) = [(F_sample_ − F_0_)/(F_triton_ − F_0_) ×100] + % re-encapsulated HPTS/DPX(1)

### 3.11. Fluidity Measurement

The measurements were performed using a microplate reader (Spark, Tecan) at 32 °C at an emission wavelength from 405 nm to 520 nm. The measurements were recorded every 10 min for one hour. The mixture was stirred after each measurement cycle for about 3 min.

Two solutions of liposomes were prepared. The first one contained liposomes stained with the LLaurdan (see above in Laurdan-stained liposomes experiment) at 62.5 µM to obtain a final concentration of 50 µM/1 µM (liposomes/LLaurdan) after the addition of the sample. The second one contained liposomes with DMSO, the same amount (50 µM µL) as in the treated ones and is used as a control. On the other hand, solutions of GD-PAMAM-3, vitamin C and VitC@GD-PAMAM-3 (56:1 VitC:dendrimer molar ratio) were prepared in ultra-pure water to obtain final concentrations of 0.02 µM, 1.11 µM and 01.13 µM), respectively, in the mixtures of the liposome sample. The VitC@GD-PAMAM-3 complex contained the same concentration of GD-PAMAM-3. The microplate wells were filled with liposome solution with or without Laurdan (100 µL). A first measurement was made before the addition of dendrimers to compare the intensity before and after the addition. Then, 25 µL of each sample was added to the liposomes with DMSO or to the Laurdan-stained liposomes. The blank one was obtained by adding the same volume of ultra-pure water (25 µL). Three replicates for each sample were performed. The measurements were recorded every 10 min for one hour. An emission spectrum was recorded from 405 to 520 nm, but only the wavelengths of 440 and 490 nm were taken into account for calculating the Generalized Polarization (GP_ex_) parameter (Equation (2)): GP_ex_ = (I_440_ − I_490_)/(I _440_ + I_490_)(2)
where I_440_ and I_490_ are the emission intensities subtracted from the blank at 440 and 490 nm, respectively. Each experiment has been repeated three times under the same conditions with a new batch of liposomes.

## 4. Conclusions

The aim of our work was to evaluate the potential of different generations of two biobased glycerodendrimer families (GlyceroDendrimers-Poly(AmidoAmine) (GD-PAMAMs) or GlyceroDendrimers-Poly(Propylene Imine) (GD-PPIs)) as vitamin C carriers for topical application. For that purpose, we first investigated their capacity to encapsulate vitamin C. The results showed that our system GD-PAMAM-3 has a higher encapsulating capacity than commercial PAMAM-3 and different GD-PPIs dendrimers. Additionally, our best candidate, GD-PAMAM-3, has no cytotoxicity on human dermal fibroblasts, whereas GD-PPIs have a cytotoxic effect at concentrations higher or equal to 0.25 mg/mL, which renders GD-PAMAM-3 another favorable property.

To better understand its mechanism of action and its use in the cosmetics, the interaction of the best candidate, GD-PAMAM-3, encapsulating vitamin C or not, with SC biomimetic models, was investigated by using two kinds of SC models, lipid monolayers and liposomes. The study with the lipid monolayer model has shown that the dendrimer encapsulating vitamin C or not is able to interact with the main lipids of the SC at all pH setups tested. However, it is suggested that only pH 5.0 is favorable for release of vitamin C into the SC matrix. Development of liposomes mimicking SC lipid composition and with an internal core at pH 7.4 and an external environment of pH 5.0 to better mimic real skin conditions was performed for investigating the effect of dendrimers on SC permeability and fluidity/rigidity. The binding of GD-PAMAM-3 or VitC@GD-PAMAM-3 to the SC-mimicking liposomes does not affect their permeability at low molar ratios of GD-PAMAM-3:liposome and has a limited permeabilizing effect at higher ratios, in accordance with the absence of cytotoxicity on dermal cells. The binding of GD-PAMAM-3 and VitC@GD-PAMAM-3 increases the rigidity of the liposome bilayer, with a higher immediate effect in the presence of VitC. It suggests a reinforcement of the SC barrier property by GD-PAMAM-3 with a more intense immediate effect in the presence of VitC.

In conclusion, our results suggest that GD-PAMAM-3 is the best dendrimer candidate for VitC encapsulation for topical application. It has high encapsulating ability, low cytotoxicity and is suggested to be able to deliver vitamin C to the SC. However, additional assays have to be performed to confirm this last property.

## Data Availability

Not applicable.

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
