# Peer review of "Encapsulation of Vitamin C by Glycerol-Derived Dendrimers, Their Interaction with Biomimetic Models of *Stratum corneum* and Their Cytotoxicity"

_molecules, 2022, doi:10.3390/molecules27228022_

Round 1

Reviewer 1 Report

This work is of great interest in the field of application of dendrimers. The encapsulation and stability of vitamin c is an extremely important problem and the authors have found an elegant solution to this problem with the help of dendrimers. It is worth noting the promise of this approach. I think that this publication can be considered for publication in its present form.

Author Response

Dear reviewer,

On behalf of my co-authors and myself, I would like to thank you and the reviewers for the comments. We sincerely appreciate the thorough analysis of the work as well as the timely fashion with which the report has been communicated to us.

We have analysed all the criticisms raised about the work and modified the manuscript to take into account all the suggestions.

The changes are marked up using the “Track Changes” function.

We hope that with these changes made, our contribution will be acceptable for publication in Molecules.

Best regards,

Magali Deleu

Reviewer 2 Report

The article is well-written and correctly presented, but I have doubts about its relevance in the field considering that: 

- The synthesis of the dendrimers is ambiguously included in the abstract, the introduction, and the conclusions as if they were made for this article. Still, the authors employ dendrimers already synthesized and reported years ago (references 13 and 15). They also studied their cytotoxicity in fibroblasts with WST1 assays in reference 13. Therefore, I am not sure if Sections 2.1. and 2.3. belong to this article, and that weakens its scientific value.

- GD-PPI dendrimers do not work better than undecorated and commercial PPI dendrimers. They are indeed discarded for the SC interaction experiments, which are then reduced to only one dendrimer: GD-PAMAM-3.

Other comments:

1. Table 3 relates to Figure 4 and I think it should say Figure 3. 

2. In my opinion, It would be valuable to compare PAMAM-3 with GD-PAMAM-3 in terms of their interaction with the SC-mimicking membranes to enforce the idea that this type of dendrimers not only encapsulates more VitC than the commercial ones but also shows more potential to interact with them. 

Author Response

Dear reviewer,

On behalf of my co-authors and myself, I would like to thank you and the reviewers for the comments. We sincerely appreciate the thorough analysis of the work as well as the timely fashion with which the report has been communicated to us.

We have analysed all the criticisms raised about the work and modified the manuscript to take into account all the suggestions.

The changes are marked up using the “Track Changes” function.

Hereafter, you will find our responses to your comments, raised point by point.

We hope that with these changes made, our contribution will be acceptable for publication in Molecules.

Best regards,

Magali Deleu

 - The synthesis of the dendrimers is ambiguously included in the abstract, the introduction, and the conclusions as if they were made for this article. Still, the authors employ dendrimers already synthesized and reported years ago (references 13 and 15). They also studied their cytotoxicity in fibroblasts with WST1 assays in reference 13. Therefore, I am not sure if Sections 2.1. and 2.3. belong to this article, and that weakens its scientific value.

The part devoted to the synthesis has been moved to the materials and methods section. Also, the emphasis on synthesis has been removed in the abstract, introduction and conclusion so as not to give the reader the impression that the methodology was developed for the present work.

However, the previously published cytotoxic study looked at MRC5 fibroblasts, which are lung cells, not dermal fibroblasts. In our article, the cytotoxic study was carried out on dermal fibroblasts to adapt to the objective of the work relating to the cosmetic application. These are therefore unpublished results never published elsewhere.

- GD-PPI dendrimers do not work better than undecorated and commercial PPI dendrimers. They are indeed discarded for the SC interaction experiments, which are then reduced to only one dendrimer: GD-PAMAM-3.

Indeed, GD-PPI dendrimers do not work better than undecorated and commercial PPI in terms of vitC encapsulation. However, their cytotoxicity is higher than the PAMAM and GD-PAMAM ones and as GD-PAMAM-3 has a higher VitC encapsulating capacity than PAMAM-3, we decided to focus only on GD-PAMAM-3 for the biophysical study of the mechanism of action.  

Other comments:

1. Table 3 relates to Figure 4 and I think it should say Figure 3.

Correction has been done.

2. In my opinion, It would be valuable to compare PAMAM-3 with GD-PAMAM-3 in terms of their interaction with the SC-mimicking membranes to enforce the idea that this type of dendrimers not only encapsulates more VitC than the commercial ones but also shows more potential to interact with them.

The higher VitC encapsulation capacity of GD-PAMAM-3 compared to PAMAM-3 make it the best candidate for vitC delivery for topical application. This is why we only focused on the GD-PAMAM-3 for its SC interaction properties in order to further investigate its mechanism of action. This is now better expressed in the manuscript (see abstract, end of introduction and end of 2.2 section).

Reviewer 3 Report

In the present study, Deleu and co-authors present an interesting research paper about the encapsulation of vitamin C using glycerol-derived dendrimers, and their interaction with skin membrane and cytotoxicity studies. The work is good for drug delivery applications. This is excellent work - well organized, thoroughly reviewed, well written, and covers the all the required work that needs to be done. The manuscript is very well presented and enjoyable to read.

There are a few minor issues that should be taken into consideration, as indicated in the following:

1.     The manuscript should clearly state the novelty of work in the abstract.

2.     The title should be changed as the authors also synthesized liposomes in the manuscript. Why you have synthesized the liposomes, it is little bit confusing as title is based on encapsulation of vitamin C in the dendrimer and you are also synthesizing liposomes, please clarify. Moreover, there is no line in the abstract about liposomes synthesis.

3.     I suggest, authors should also go for FTIR interaction studies to check the incompatibilities if any.

4.     Kindly reduce the number of abbreviations in the whole manuscript.

5.     Please add some latest articles of year 2021 and 2022.

6.     References needs to be checked as per journals requirements.

7.     Please add some studies related to transdermal delivery through dendrimer, polymer-based approaches.

8.     Please update the future scope of synthesized dendrimers for drug delivery applications.

9.     Moreover, also the addition of very recent papers is desirable in the introduction as well as in results and discussion part. You can add from the following

·        Asmita Deka Dey, Ashkan Bigham, Yasaman Esmaeili, Milad Ashrafizadeh, Farnaz Dabbagh Moghaddam, Shing Cheng Tan, Satar Yousefiasl, Saurav Sharma, Ana Cláudia Paiva-Santos, Aziz Maleki, Navid Rabiee, Alan Prem Kumar, Vijay Kumar Thakur, Gorka Orive, Esmaeel Sharifi, Arun Kumar, Pooyan Makvandi. Dendrimers as nanoscale vectors: Unlocking the bars of cancer therapy, Seminars in Cancer Biology, 2022

·        Sakineh Hajebi, Satar Yousefiasl, Ilnaz Rahimmanesh, Alireza Dahim, Sepideh Ahmadi, Firoz Babu Kadumudi, Nikta Rahgozar, Sanaz Amani, Arun Kumar, Ehsan Kamrani, Mohammad Rabiee, ssunta Borzacchiello, Xiangdong Wang, Navid Rabiee, Alireza Dolatshahi-Pirouz, Pooyan Makvandi, Genetically engineered viral vectors and organic-based non-viral nanocarriers for drug delivery applications, Advanced Healthcare Materials, (2022).

·        Mahnaz Hassanpour, Mohammad Hassan Shahavi, Golnaz Heidari, Arun Kumar, Mehrab Nodehi, Farnaz Dabbagh Moghaddam, Mahsa Mohammadi, Nasser Nikfarjam, Esmaeel Sharifi, Pooyan Makvandi, Hasan Karimi Malek, Ehsan Nazarzadeh Zare, Ionic liquid-mediated synthesis of metal nanostructures: Potential application in cancer diagnosis and therapy. Journal of Ionic Liquids, 2(2), 2022, 100033.

·        Asmita Deka De, Arun Kumar, Satar Yousefiasl, Farnaz Dabbagh Moghaddam, Ilnaz Rahimmanesh, Mohamadmahdi Samandari, Sumit Jamwal, Aziz Maleki, Abbas Mohammadi, Navid Rabiee, Ebrahim Mostafavi, Ali Tamayol, Esmaeel Sharifi*, Pooyan Makvandi*, miRNA-encapsulated abiotic materials and biovectors for cutaneous and oral wound healing: biogenesis, mechanisms, and delivery nanocarriers. Bioengineering and Translational Medicine

Author Response

Dear reviewer,

On behalf of my co-authors and myself, I would like to thank you and the reviewers for the comments. We sincerely appreciate the thorough analysis of the work as well as the timely fashion with which the report has been communicated to us.

We have analysed all the criticisms raised about the work and modified the manuscript to take into account all the suggestions.

The changes are marked up using the “Track Changes” function.

Hereafter, you will find our responses to your comments, raised point by point.

We hope that with these changes made, our contribution will be acceptable for publication in Molecules.

Best regards,

Magali Deleu

1. The manuscript should clearly state the novelty of work in the abstract.

The abstract has been revised accordingly.

2. The title should be changed as the authors also synthesized liposomes in the manuscript. Why you have synthesized the liposomes, it is little bit confusing as title is based on encapsulation of vitamin C in the dendrimer and you are also synthesizing liposomes, please clarify. Moreover, there is no line in the abstract about liposomes synthesis.

We realized that the liposome formulation section was a little bit confusing for the reader.

Liposome formulation was developed to perform the biophysical study of the dendrimer-SC interaction but not for encapsulating VitC.

So, the text has been change in different places to better express this (see the abstract, end of introduction and beginning of 2.3.2 section).

Moreover, we have added in supplementray data, cryo-TEM images for liposomes in absence or presence of dendrimer to prove that vesicles are well formed and that they are not damaged by the dendrimer but only aggregated, in accordance with the absence or limited leakage observed on liposomes by the HPTS/DPX leakage assays.

The title has also been revised to avoid this confusion.

3. I suggest, authors should also go for FTIR interaction studies to check the incompatibilities if any.

FTIR could be a valuable technique to characterize the interaction between VitC and the dendrimer but as we have not the technique available in the lab we have preferred to use NMR technology to characterize the encapsulation of VitC within the dendrimer.

4. Kindly reduce the number of abbreviations in the whole manuscript.

To make the reading easier, a list of abbreviations has been added after the conclusion section.

5. Please add some latest articles of year 2021 and 2022.

Several references of year 2021 and 2022 have added in the introduction.

6. References needs to be checked as per journals requirements.

References were all checked and corrected when needed to fit with the journal requirements.

7. Please add some studies related to transdermal delivery through dendrimer, polymer-based approaches.

Two recent reviews highlighted the interest of nanocarriers for transdermal drug delivery were added in the introduction.

8. Please update the future scope of synthesized dendrimers for drug delivery applications.

Some sentences were added in the beginning of the introduction.

9. Moreover, also the addition of very recent papers is desirable in the introduction as well as in results and discussion part. You can add from the following.

Some of them were added in the introduction.

Round 2

Reviewer 2 Report

None